# Gender differences in trunk appearance perception and health-related quality of life (HRQoL) in the patients with moderate adolescent idiopathic scoliosis (AIS) undergoing orthotic treatment: An observational study

**JoJo Yiying ZOU**, **Tung LI, Man Sang WONG**\*

Department of Biomedical Engineering, The Hong Kong Polytechnic University, Hong Kong SAR, China

\* m.s.wong@polyu.edu.hk

## Abstract

### Background

Orthotic treatment is commonly used as a non-surgical intervention for managing moderate adolescent idiopathic scoliosis (AIS). Although prior studies have evaluated various factors influencing health-related quality of life (HRQoL) in the patients with AIS, the association between trunk appearance perception and HRQoL, including potential gender differences, remains insufficiently defined. This study investigated gender differences in trunk appearance perception and its relationship with QoL among the patients with moderate AIS undergoing orthotic treatment.

### Methods

Patients with moderate AIS undergoing orthotic treatment were included, and HRQoL evaluations were conducted before treatment, as well as one and seven months after the initiation of orthotic treatment. The evaluation tools included the Trunk Appearance Perception Scale (TAPS), Scoliosis Research Society-22r (SRS-22r), and Brace Questionnaire (BrQ). The study time points were selected to capture baseline conditions, initial adaptation, and longer-term treatment effects.

### Results

A total of 34 females and 11 males participated in the study. No significant gender differences were observed in global HRQoL, with both groups consistently reporting low self-image and self-esteem. In females, increasing age was negatively correlated with TAPS scores, whereas in males, it showed a positive correlation. For females, higher compliance was associated with higher bodily pain scores on the BrQ ($\rho = 0.417$), indicating reduced pain levels. In contrast, among males, greater compliance was

**Data availability statement:** All relevant data are within the paper and its Supporting information file.

**Funding:** The author(s) received no specific funding for this work.

**Competing interests:** The authors have declared that no competing interests exist.

associated with poor trunk appearance perception after seven months ($\rho = -0.619$). While TAPS was unrelated to SRS-22r and BrQ in females, a more favorable trunk appearance in males was strongly related to better function and self-image scores on the SRS-22r after seven months ($\rho = 0.614$ and $0.703$, respectively).

## Conclusion

Trunk appearance perception and overall HRQoL were similar between females and males in this study. However, the score related to self-image was lower than other HRQoL domains.

## Introduction

Adolescent idiopathic scoliosis (AIS) is a complex spinal deformity affecting approximately 1–3% of adolescents, characterized by a lateral curvature of the spine accompanied by vertebral rotation [1]. AIS can lead to noticeable changes in trunk appearance and increase the risk of experiencing back pain [2]. In severe cases, the progressive deformity may result in significant disability and diminish the health-related quality of life (HRQoL) [3], making control of curve progression essential. For moderate AIS, orthotic treatment is commonly prescribed to prevent further deterioration and avoid the need for spinal surgery [4,5]. However, the success of orthotic treatment depends not only on its mechanical effectiveness but also on patient compliance, which can be challenging due to the physical discomfort, altered body appearance, and restrictions on daily activities imposed by orthosis wear [6,7]. As the orthosis is a visible and constant reminder of the condition, its negative effects on self-perception and HRQoL can further complicate compliance with treatment protocols.

HRQoL is a multidimensional concept that includes physical health, functional ability, emotional well-being, and social relationships [8–10]. In the patients with moderate AIS, both spinal deformity and long-term orthotic treatment can lead to psychosocial challenges [11–13]. In particular, trunk appearance perception, how people subjectively evaluate the aesthetic qualities of their torso, plays a key role in overall body image. Changes in trunk appearance can affect self-esteem and emotional health, thereby influencing social interactions and overall HRQoL [14]. Moreover, prolonged orthosis use can alter lifestyle and increase stress, anxiety, and depression, further reducing self-esteem and leading to a decline in HRQoL [15–17]. These findings underscore the importance of evaluating not only the physical but also the psychological impacts of orthotic treatments for AIS.

The HRQoL and its associated factors in patients with AIS undergoing orthotic treatment have been investigated in current studies. One longitudinal study demonstrated that females maintained stable assessments of body function and mental health, and showed significant improvements in trunk shape perception after 12 months. A negative perception of self-image coexisted with severe emotional distress [17]. Studies in the orthotic treatment found that males exhibited better mental health

outcomes and a higher perception of body image compared to females [11,18]. In contrast, additional investigations have reported no significant gender differences in HRQoL outcomes among patients with orthotic treatment [13,19]. Despite these efforts, few studies have evaluated the changes in HRQoL during the orthotic treatment and the related clinical and radiological parameters in males versus females with AIS.

The present study aimed to investigate gender differences in trunk appearance perception and HRQoL among adolescents with moderate AIS undergoing orthotic treatment at different stages of therapy. Additionally, we examined the correlations between relevant clinical parameters, trunk appearance perception, and HRQoL between females and males. We hypothesized that HRQoL would differ between genders and that the factors influencing these outcomes might vary throughout the treatment course.

## Materials and methods

### Participants

The patients with moderate AIS conservatively treated with thoracolumbosacral orthosis (TLSO) were included in the study. These patients were recruited from a local spine care community project. Eligibility criteria included: 1) age between 10 and 15 years; 2) moderate AIS with a major Cobb angle of 20–45 degrees; and 3) prescribed orthotic treatment. Exclusion criteria were: 1) prior AIS treatment; 2) reluctance to have orthotic treatment; and 3) inability to complete the full treatment course during the study period. Informed written consent was obtained from all patients and their parents before participation. The selection period lasted 22 months, from March 2023 to December 2024. This study was approved by the Hong Kong Polytechnic University Institutional Review Board (HSEARS20221012006).

Each patient was prescribed a custom-fabricated Hong Kong style TLSO (Fig 1) by a qualified orthotist and instructed to wear it 23 hours a day. The orthosis was equipped with strap markings to ensure correct tightness. All patients received the same type of orthosis, thereby standardizing the treatment and minimizing potential variations in outcomes attributable to differences in orthosis design. A pre-study power analysis for the Mann–Whitney U test ($\alpha = 0.05$, 80% power, and a superiority probability of 0.68 based on clinical and literature considerations [20]) indicated that 42 participants were

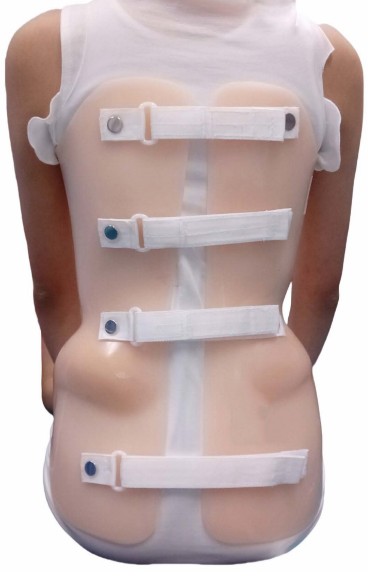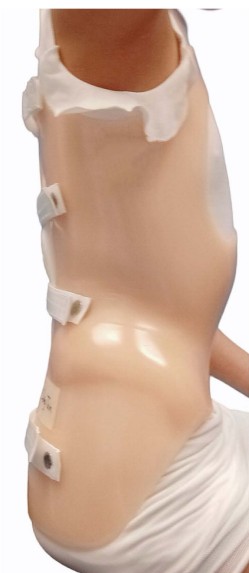

**Fig 1. The Hong Kong style TLSO (frontal and sagittal view).**

required. Initially, 48 patients with moderate AIS were eligible for the study. However, two participants withdrew after one month due to discomfort from the orthosis, and one participant only wore the orthosis for around 2 hours per day, which may not have provided sufficient exposure to the orthotic treatment [21].

## Evaluation instruments

Three questionnaires were used in this study, with both English and Chinese versions available for patients in a bilingual society. This dual-language approach facilitates direct comparisons with international studies and ensures that participants can select their preferred language, enhancing comprehension and data quality. It typically took patients 18–25 minutes to complete all three questionnaires.

The Trunk Appearance Perception Scale (TAPS) (Fig 2) (S1 File) was used to assess trunk deformity perception in the patients with AIS. The TAPS includes three sets of five figures depicting the trunk from different angles: back view, head view (with the patient bending forward), and front view. The inclusion of both the back view and the forward-bending view is for capturing how patients believe others perceive their trunk appearance, a key aspect of self-image in scoliosis [22]. Scores for each set range from 1 (greatest deformity) to 5 (smallest deformity), with higher scores indicating a more positive perception of trunk appearance. The TAPS demonstrates excellent internal consistency (Cronbach's $\alpha = 0.89$) and test-retest reliability (intraclass correlation coefficient (ICC) = 0.92) [22]. Similarly, the Chinese-translated TAPS has been validated in prior studies, showing acceptable internal consistency (Cronbach's $\alpha = 0.70$) and reproducibility (ICC = 0.95) [23].

The Scoliosis Research Society-22r (SRS-22r) questionnaire (S2 File), recommended by the Scoliosis Research Society for evaluating treatment outcomes, has been widely used in previous research as a comprehensive measure of overall HRQoL [11,24–26]. This questionnaire consists of 22 questions across five domains: function, pain, self-image, mental health, and treatment satisfaction. Each question is scored from 1 (worst) to 5 (best), with domain and total scores expressed as averages, ranging from 1 to 5. Higher scores indicate better HRQoL. The SRS-22r has demonstrated internal consistency coefficients ranging from 0.75 to 0.92 and reproducibility from 0.85 to 0.96 [27]. Moreover, the Chinese-translated SRS-22 questionnaire has shown excellent test-retest reproducibility, with an ICC of 0.75 across all domains [28].

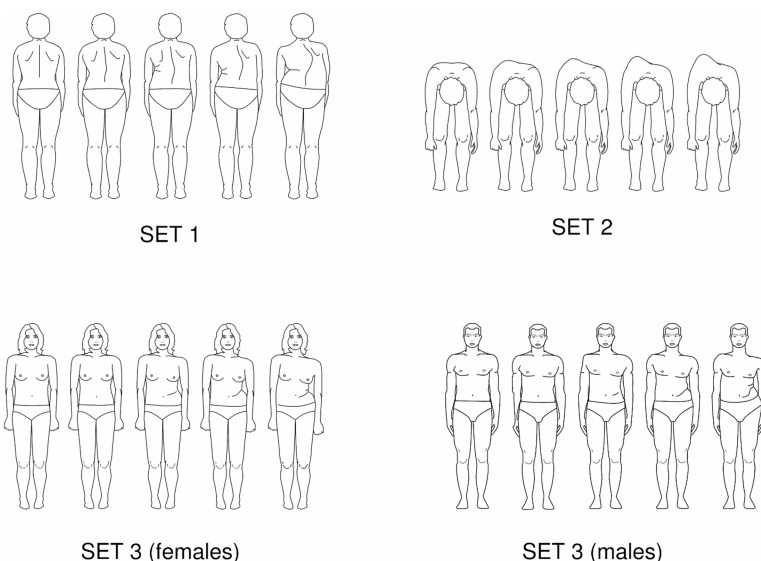

**Fig 2. Trunk Appearance Perception Scale (TAPS).**

The Brace questionnaire (BrQ) (S3 File) is a disease-specific tool for evaluating HRQoL in the patients with AIS undergoing orthotic treatment. The BrQ contains 34 questions across eight domains: general health perception, physical functioning, emotional functioning, self-esteem and aesthetics, vitality, school activity, bodily pain, and social functioning. Each item is rated on a 5-point Likert scale from 1 (poorest) to 5 (best). The total BrQ score is obtained by summing all 34 item scores, multiplying the sum by 20, and dividing by 34, yielding a normalized score between 20 and 100 (with higher scores indicating better HRQoL). A subscale score for each domain is calculated by dividing the domain's total score by the number of items in that domain. Patients also report the orthotic treatment start date and average daily wear time. The original BrQ demonstrates satisfactory internal consistency reliability (Cronbach's α = 0.82) [29] and the Chinese-translated version has shown very good internal consistency (Cronbach's α = 0.89) and excellent test-retest reproducibility (ICC = 0.83) [30].

### Evaluations

This prospective study included routine clinical radiological assessments and three questionnaire evaluations conducted during clinical visits. A baseline radiological assessment was conducted as part of the standard diagnostic procedure for scoliosis. To evaluate the corrective effect of the TLSO, the in-brace X-ray was obtained one month after the initial orthosis fitting. This timing was selected to allow patients sufficient time to adapt to the orthosis. Furthermore, the patients were required to wear the brace continuously for a full day before imaging to ensure that the measured correction accurately reflected its maximal effect on spinal deformities [31]. The major spinal curvature was measured using the Cobb angle, based on the two most tilted vertebrae [32]. The in-brace correction rate was calculated by subtracting the in-brace Cobb angle from the baseline Cobb angle and then dividing the result by the baseline Cobb angle.

The questionnaire evaluation timeline was based on clinical practice and some previous studies. An investigator provided explanations or clarifications as necessary. The first evaluation, conducted before orthosis fitting, employed only the TAPS to capture baseline trunk appearance perception [33,34]. One month after treatment initiation, an integrated questionnaire (TAPS, SRS-22r, and BrQ) was administered to assess early adaptation. This one-month follow-up was strategically chosen because the initial orthotic treatment period is critical for long-term outcomes [35], while also aligning with standard clinical protocols that allow sufficient time for patient adaptation yet enable early orthosis adjustments if needed. The third evaluation was originally scheduled for six months after the second assessment (i.e., seven months post-orthosis fitting) to capture longer-term treatment effects, in accordance with both the 2018 SOSORT guidelines recommending 3–6 months follow-ups [36] and protocols from previous studies using 6-month intervals [17,33]. By the third evaluation, the patients were informed of their in-brace correction results. However, because of disruptions caused by the COVID-19 pandemic, some follow-up sessions were rescheduled to ensure participant safety and data collection, resulting in the third evaluation being completed between 6 and 17 months after having orthotic treatment, with an average follow-up period of 10.0 months.

### Statistical analyses

Statistical analyses were performed using IBM SPSS version 27.0 (IBM Corp., Armonk, NY, USA). Descriptive statistics, including means and standard deviations, were used to characterize the distribution of quantitative variables. Data normality was assessed using the Kolmogorov-Smirnov test. An independent samples $t$-test was used to compare age, baseline major Cobb angle, in-brace correction rate, and orthotic treatment compliance at the 2nd and the 3rd evaluations, and the total treatment period across genders. The Mann-Whitney U test was applied to evaluate gender differences in questionnaire responses. A significance threshold of $p < 0.05$ was used. A Friedman two-way analysis of variance (ANOVA) by ranks was conducted to assess differences in TAPS scores across the three time points, with significant results further explored using multiple Wilcoxon signed-rank tests. Spearman's rank-order correlation coefficients (ρ) were calculated to examine correlations between TAPS, SRS-22r, and BrQ results and clinical characteristics for both genders. Correlations were classified as strong (ρ > 0.60), moderate (0.30 ≤ ρ ≤ 0.60), or weak (ρ < 0.30) [37].

## Results

### Demographic analysis

A total of 45 participants (34 females and 11 males) completed the study. The demographic characteristics are detailed in Table 1. Statistical analysis revealed no significant differences ($p > 0.05$) between females and males regarding age, baseline Cobb angle, in-brace correction rate, orthosis wearing compliance at the 2nd and the 3rd evaluations, and the total duration of orthotic treatment in this study.

### Questionnaires results

Fig 3 presents the mean scores for females and males obtained from three questionnaires at three different evaluations. No significant gender differences were observed in any questionnaire at any evaluation, and the scores remained stable

**Table 1. Demographic Analysis.**

| Patients' characteristics | Females (n = 34) (Mean ± SD) | Males (n = 11) (Mean ± SD) | Sig. |
|---|---|---|---|
| **Age (year)** | 11.8 ± 1.3 | 12.4 ± 1.4 | 0.255 |
| **Baseline Cobb angle (°)** | 28.3 ± 6.8 | 27.1 ± 4.9 | 0.765 |
| **In-brace correction rate (%)** | 52.8 ± 28.8 | 60.6 ± 28.1 | 0.213 |
| **Compliance at the 2nd Evaluation (hours/day)** | 15.1 ± 6.7 | 12.8 ± 6.9 | 0.331 |
| **Compliance at the 3rd Evaluation (hours/day)** | 13.3 ± 6.0 | 11.2 ± 5.8 | 0.323 |
| **Treatment Duration (months)** | 9.6 ± 3.4 | 9.3 ± 2.6 | 0.757 |

SD: standard deviation

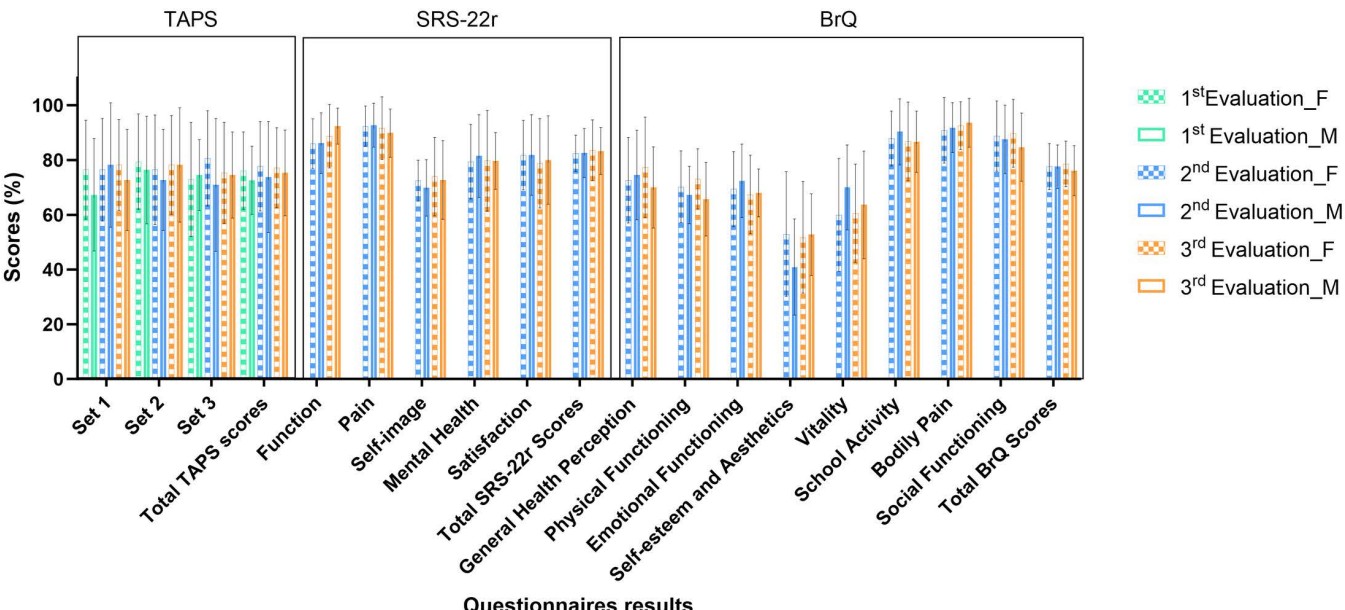

**Fig 3. Mean scores for the TAPS, SRS-22r, and BrQ questionnaires over three evaluations by gender (F: female; M: male).** The error bars indicate the standard deviation of the mean scores.

over time for both genders. Additionally, in the SRS-22r and BrQ questionnaires, the domains related to self-image, self-esteem and aesthetics consistently recorded the lowest scores compared to other HRQoL domains.

## Correlation of questionnaire results with baseline and treatment parameters

Table 2 shows the correlation analysis between TAPS scores and different baseline and treatment parameters for both genders. In females, age showed moderate negative correlations with TAPS scores in set 1, set 3 and total score ($\rho = -0.368$, $-0.397$, and $-0.345$, respectively). The baseline Cobb angle was moderately and negatively correlated with TAPS set 1 and the total score ($\rho = -0.353$ and $-0.368$). In males, age demonstrated strong positive correlations with TAPS scores in set 1, set 2 and total score ($\rho = 0.614$, $0.673$, and $0.772$, respectively). Compliance at the third evaluation exhibited strong negative correlations with TAPS set 2, set 3, and the total score ($\rho = -0.638$, $-0.637$, and $-0.619$, respectively).

Table 3 summarizes the correlation analysis between SRS-22r scores and various baseline and treatment parameters for both genders. In females, a significant negative correlation was found between treatment duration and pain scores at the third evaluation ($\rho = -0.356$). In contrast, no significant correlations were detected between any of the parameters and SRS-22r scores in males at either evalutaion period.

Table 4 presents the correlation analysis between BrQ scores and baseline as well as treatment parameters for females and males. In females, a negative correlation was found between age and emotional functioning ($\rho = -0.356$).

**Table 2. Correlation of TAPS with the baseline and treatment parameters.**

| Gender | Baseline & treatment parameters | Evaluation | TAPS | | | |
|---|---|---|---|---|---|---|
| | | | Set 1 ρ (p-value) | Set 2 ρ (p-value) | Set 3 ρ (p-value) | Total ρ (p-value) |
| **Female (n = 34)** | Age | 1st | −0.368* (0.032) | −0.184 (0.298) | −0.397* (0.020) | −0.345* (0.046) |
| | Baseline Cobb angle | 1st | −0.353* (0.041) | −0.305 (0.079) | −0.220 (0.211) | −0.368* (0.032) |
| | In-brace correction | 3rd | −0.110 (0.536) | −0.143 (0.420) | −0.166 (0.349) | −0.148 (0.405) |
| | Compliance at the 2nd Evaluation | 2nd | −0.022 (0.900) | −0.090 (0.611) | −0.293 (0.092) | −0.143 (0.419) |
| | Compliance at the 3rd Evaluation | 3rd | 0.099 (0.577) | 0.119 (0.501) | −0.009 (0.960) | 0.097 (0.585) |
| | Treatment Duration | 3rd | 0.119 (0.504) | −0.155 (0.382) | 0.057 (0.750) | 0.031 (0.862) |
| **Male (n = 11)** | Age | 1st | 0.614* (0.045) | 0.673* (0.023) | 0.566 (0.069) | 0.772* (0.005) |
| | Baseline Cobb angle | 1st | 0.002 (0.994) | −0.426 (0.192) | 0.194 (0.567) | −0.089 (0.795) |
| | In-brace correction | 3rd | −0.029 (0.933) | −0.109 (0.749) | −0.044 (0.897) | −0.037 (0.915) |
| | Compliance at the 2nd Evaluation | 2nd | −0.367 (0.266) | 0.075 (0.828) | 0.270 (0.423) | −0.051 (0.881) |
| | Compliance at the 3rd Evaluation | 3rd | −0.407 (0.215) | −0.638* (0.035) | −0.637* (0.035) | −0.619* (0.042) |
| | Treatment Duration | 3rd | −0.272 (0.418) | −0.122 (0.722) | −0.317 (0.342) | −0.278 (0.408) |

ρ, spearman correlation coefficient;

*Correlation significant at $p < 0.05$.

**Table 3. Correlation between SRS-22r and the baseline and treatment parameters.**

| Baseline & treatment parameters | Evaluation | SRS-22r | | | | | |
|---|---|---|---|---|---|---|---|
| | | Function ρ (p-value) | Pain ρ (p-value) | Self-image ρ (p-value) | Mental health ρ (p-value) | Satisfaction ρ (p-value) | Total ρ (p-value) |
| **Female (n = 34)** | | | | | | | |
| Age | 2nd | −0.155 (0.380) | −0.131 (0.459) | −0.167 (0.347) | −0.038 (0.830) | −0.310 (0.074) | −0.233 (0.184) |
| Baseline Cobb Angle | 2nd | −0.213 (0.226) | −0.140 (0.429) | 0.297 (0.088) | −0.164 (0.353) | −0.163 (0.357) | −0.190 (0.281) |
| In-brace Correction | 3rd | −0.321 (0.065) | −0.247 (0.159) | 0.025 (0.889) | 0.096 (0.590) | 0.102 (0.565) | 0.011 (0.949) |
| Compliance at the 2nd Evaluation | 2nd | −0.113 (0.526) | −0.078 (0.660) | 0.032 (0.858) | −0.238 (0.176) | 0.218 (0.216) | −0.182 (0.303) |
| Compliance at the 3rd Evaluation | 3rd | −0.034 (0.847) | −0.043 (0.811) | 0.137 (0.440) | −0.220 (0.212) | −0.104 (0.560) | −0.027 (0.881) |
| Treatment duration | 3rd | −0.241 (0.170) | −0.356* (0.039) | −0.007 (0.970) | 0.140 (0.429) | 0.126 (0.476) | −0.027 (0.881) |
| **Male (n = 11)** | | | | | | | |
| Age | 2nd | 0.418 (0.201) | 0.464 (0.150) | 0.355 (0.283) | 0.019 (0.956) | 0.086 (0.801) | 0.331 (0.320) |
| Baseline Cobb Angle | 2nd | 0.046 (0.892) | −0.164 (0.631) | 0.092 (0.787) | 0.162 (0.635) | 0.191 (0.573) | 0.088 (0.798) |
| In-brace Correction | 3rd | −0.357 (0.281) | −0.332 (0.319) | −0.201 (0.554) | −0.545 (0.083) | 0.028 (0.936) | −0.391 (0.235) |
| Compliance at the 2nd Evaluation | 2nd | 0.097 (0.776) | −0.258 (0.443) | −0.046 (0.892) | −0.356 (0.282) | 0.241 (0.476) | −0.028 (0.936) |
| Compliance at the 3rd Evaluation | 3rd | −0.554 (0.077) | −0.306 (0.360) | −0.563 (0.071) | −0.339 (0.308) | 0.295 (0.378) | −0.275 (0.412) |
| Treatment duration | 3rd | −0.129 (0.706) | 0.014 (0.967) | −0.377 (0.253) | −0.254 (0.452) | −0.209 (0.538) | −0.285 (0.395) |

ρ, spearman correlation coefficient;

*Correlation significant at p < 0.05.

Social functioning showed a negative correlation with in-brace correction at the third evaluation (ρ = −0.360), while bodily pain demonstrated a positive correlation with compliance at the second evaluation (ρ = 0.417). In males, no significant correlations were detected between any BrQ domains and the baseline or treatment parameters.

### Associations between TAPS, SRS-22r, and BrQ results

Table 5 displays the associations among TAPS, SRS-22r, and BrQ outcomes for both females and males. In females, no significant associations were observed for either the SRS-22r or BrQ scores at any evaluation. However, in males, strong positive correlations were identified at the third evaluation between the self-image and function domains of the SRS-22r questionnaire and the TAPS total score (ρ = 0.614 and ρ = 0.703, respectively). Notably, when considering the overall TAPS and SRS-22r responses in males at the second evaluation, no significant association was found.

### Discussion

The study investigated the trunk appearance perception and HRQoL, as measured by TAPS, SRS-22r, and BrQ, in patients undergoing orthotic treatment for AIS across genders. No significant gender differences were observed in

**Table 4. Correlation between the BrQ and the baseline and treatment parameters.**

| Baseline & treatment parameters | Evaluation | BrQ | | | | | | | | |
|---|---|---|---|---|---|---|---|---|---|---|
| | | General health perception ρ (p-value) | Physical functioning ρ (p-value) | Emotional functioning ρ (p-value) | Self-esteem and aesthetics ρ (p-value) | Vitality ρ (p-value) | School Activity ρ (p-value) | Bodily pain ρ (p-value) | Social functioning ρ (p-value) | Total score ρ (p-value) |
| **Female (n = 34)** | | | | | | | | | | |
| Age | 2nd | −0.028 (0.877) | −0.250 (0.153) | −0.356* (0.039) | −0.198 (0.261) | −0.147 (0.408) | −0.389* (0.023) | −0.016 (0.927) | −0.107 (0.545) | −0.317 (0.068) |
| Baseline Cobb Angle | 2nd | 0.252 (0.150) | 0.038 (0.830) | 0.172 (0.330) | −0.048 (0.787) | 0.294 (0.091) | 0.120 (0.498) | −0.094 (0.598) | 0.001 (0.996) | 0.189 (0.284) |
| In-brace Correction | 3rd | −0.195 (0.270) | −0.085 (0.631) | 0.024 (0.892) | −0.054 (0.760) | −0.156 (0.378) | −0.338 (0.050) | −0.165 (0.350) | −0.360* (0.037) | −0.184 (0.296) |
| Compliance (2nd) | 2nd | −0.219 (0.213) | 0.221 (0.209) | −0.195 (0.269) | −0.138 (0.438) | −0.091 (0.608) | 0.067 (0.705) | 0.417* (0.014) | −0.020 (0.909) | 0.022 (0.904) |
| Compliance (3rd) | 3rd | 0.024 (0.891) | 0.149 (0.401) | 0.091 (0.610) | 0.055 (0.756) | −0.196 (0.267) | −0.103 (0.564) | 0.083 (0.639) | 0.097 (0.586) | 0.075 (0.675) |
| Treatment duration | 3rd | −0.219 (0.214) | −0.136 (0.442) | 0.205 (0.245) | 0.268 (0.125) | 0.203 (0.250) | −0.078 (0.662) | −0.180 (0.308) | −0.088 (0.623) | 0.095 (0.595) |
| **Male (n = 11)** | | | | | | | | | | |
| Age | 2nd | −0.269 (0.423) | −0.103 (0.762) | −0.379 (0.250) | 0.344 (0.301) | −0.077 (0.822) | −0.374 (0.258) | −0.149 (0.661) | −0.591 (0.056) | −0.355 (0.284) |
| Baseline Cobb Angle | 2nd | 0.131 (0.701) | 0.572 (0.066) | 0.222 (0.512) | −0.130 (0.703) | 0.110 (0.747) | 0.354 (0.285) | 0.160 (0.639) | 0.347 (0.296) | 0.392 (0.233) |
| In-brace Correction | 3rd | −0.334 (0.316) | 0.156 (0.648) | −0.165 (0.627) | −0.121 (0.724) | −0.484 (0.132) | 0.079 (0.817) | 0.139 (0.684) | 0.050 (0.883) | 0.009 (0.979) |
| Compliance (2nd) | 2nd | −0.106 (0.757) | 0.349 (0.293) | 0.370 (0.262) | −0.159 (0.642) | −0.087 (0.799) | 0.173 (0.611) | 0.066 (0.847) | 0.362 (0.274) | 0.311 (0.353) |
| Compliance (3rd) | 3rd | −0.402 (0.220) | 0.286 (0.394) | −0.166 (0.625) | −0.049 (0.886) | −0.118 (0.729) | 0.007 (0.984) | 0.052 (0.878) | 0.039 (0.909) | −0.009 (0.979) |
| Treatment duration | 3rd | 0.022 (0.950) | −0.272 (0.418) | −0.494 (0.122) | −0.488 (0.128) | −0.187 (0.582) | 0.098 (0.775) | −0.356 (0.283) | −0.172 (0.614) | −0.280 (0.405) |

ρ, spearman correlation coefficient; Compliance (2nd), Compliance at the 2nd Evaluation; Compliance (3rd), Compliance at the 3rd Evaluation;

*Correlation significant at p < 0.05.

the overall scores of TAPS, SRS-22r, and BrQ at any evaluation point. Both females and males consistently showed the lowest ratings in the self-image domain of the SRS-22r (70% − 76% of the maximum score) and the self-esteem domain of the BrQ (41% − 53% of the maximum score) compared to all other subdomains. These findings align with previous research indicating that patients undergoing orthotic treatment for AIS tend to have diminished self-image [11, 13, 17, 38–41]. Therefore, enhancing trunk appearance perception is crucial for improving overall HRQoL in all the patients with AIS.

### Influence of age, baseline cobb angle, and in-brace correction on trunk appearance perception and HRQoL

In females, TAPS scores (set 1, set 3, and total score) demonstrated moderate negative correlations with age, contrasting with strong positive correlations observed in males (set 1, set 2, and total score). The TAPS scale assesses trunk appearance through three perspectives: the back view (set 1), forward-bending view (set 2) reflecting perceived external evaluation, and front view (set 3) representing self-observed appearance. These findings suggest that progressively

**Table 5. Associations between TAPS, SRS-22r, and BrQ results.**

| | | TAPS (2nd evaluation) ρ (p-value) | | TAPS (3rd evaluation) ρ (p-value) | |
| --- | --- | --- | --- | --- | --- |
| | | Female (n = 34) | Male (n = 11) | Female (n = 34) | Male (n = 11) |
| **SRS-22r** | Function | 0.133 (0.454) | 0.485 (0.131) | 0.052 (0.768) | 0.614* (0.045) |
| | Pain | 0.107 (0.547) | 0.230 (0.496) | −0.185 (0.296) | 0.226 (0.505) |
| | Self-image | 0.202 (0.253) | 0.563 (0.071) | 0.011 (0.949) | 0.703* (0.016) |
| | Mental health | −0.016 (0.930) | 0.028 (0.934) | −0.108 (0.544) | 0.430 (0.187) |
| | Satisfaction | 0.303 (0.082) | 0.336 (0.312) | 0.047 (0.793) | −0.114 (0.738) |
| | Total scores | 0.249 (0.155) | 0.412 (0.208) | −0.021 (0.907) | 0.422 (0.196) |
| **BrQ** | General health perception | −0.023 (0.897) | −0.490 (0.126) | −0.163 (0.358) | 0.111 (0.744) |
| | Physical functioning | 0.067 (0.707) | −0.065 (0.849) | −0.043 (0.811) | −0.277 (0.409) |
| | Emotional functioning | −0.039 (0.827) | −0.207 (0.542) | 0.019 (0.916) | 0.347 (0.295) |
| | Self-esteem and aesthetics | −0.068 (0.703) | 0.582 (0.061) | 0.255 (0.145) | 0.155 (0.650) |
| | Vitality | −0.109 (0.539) | 0.252 (0.454) | −0.013 (0.943) | −0.156 (0.647) |
| | School activity | 0.043 (0.810) | −0.385 (0.243) | 0.147 (0.406) | −0.272 (0.418) |
| | Bodily pain | 0.033 (0.852) | −0.186 (0.584) | 0.042 (0.815) | 0.195 (0.565) |
| | Social functioning | 0.051 (0.777) | −0.504 (0.114) | 0.035 (0.844) | −0.035 (0.919) |
| | Total scores | −0.023 (0.895) | −0.269 (0.425) | 0.004 (0.983) | 0.046 (0.893) |

*Significant at $p < 0.05$.

negative self-perception associates with advancing age in female patients, while male patients develop more favorable assessments of their trunk appearance over time. Female patients additionally exhibited moderate negative correlations between age and specific BrQ domains, particularly emotional functioning and school activity. These results corroborate existing literature demonstrating gender differences in HRQoL progression during adolescence. A large-scale study of 1,519 patients with idiopathic scoliosis reported superior HRQoL outcomes in males aged ≤19 years compared to females [11], while population studies have documented weak negative correlations between age and SRS-22r scores in healthy adolescents [42]. The observed pattern aligns with established developmental trajectories where gender disparities in subjective health perceptions typically emerge around age 12, with female adolescents consistently reporting poorer HRQoL outcomes [43].

The current findings further extend previous research highlighting the predominant influence of psychological factors over radiographic parameters on HRQoL in females with AIS [14,44]. Our study found that this pattern extends to males as well, with neither baseline Cobb angle nor in-brace correction rates demonstrating significant correlations

with HRQoL measures. Conversely, in female patients, moderate negative correlations were detected between the baseline Cobb angle and TAPS scores (set 1 and the total score), and a negative correlation emerged between in-brace correction and BrQ social functioning. While no significant gender differences in Cobb angles were identified, the marginally larger baseline curves observed in female patients (including two cases exceeding 40°) may partially explain the significant correlations between Cobb angle and TAPS outcomes ($\rho = -0.353$ to $-0.368$). This observation aligns with previous validation studies establishing TAPS as an effective discriminator for surgical candidates (curve > 45°) and those treatable by other means, having demonstrated strong negative correlations ($\rho = -0.55$) with curve severity [22].

### The relationship between compliance, treatment duration, trunk appearance perception, and HRQoL

For female patients, brace compliance during the first month showed moderate positive correlation with the BrQ bodily pain domain. This finding partially corroborate previous studies reporting improved HRQoL with better brace adherence in female patients, particularly in vitality and physical, emotional, and social functioning [14,44]. In females, a moderate negative correlation was also observed between treatment duration and SRS-22r pain score, consistent with reports of pain increased in good-compliance groups (>75% wear time) over extended periods [45]. While the referenced study included nearly half part-time orthosis users, in this study, females maintained averaged 13.3 hours of daily brace wear, comparable to their high-adherence group.

Male patients exhibited strong negative correlations between third-evaluation compliance and TAPS scores (sets 2, 3, and total), indicating better compliance associated with poorer body image perception. This aligns with findings that compliant patients maintained lower self-image scores throughout treatment [45], possibly because prolonged orthotic use intensifies body scrutiny and aesthetic dissatisfaction, as evidenced by greater trunk appearance dissatisfaction in braced versus untreated or surgically treated patients [46]. While no significant correlations were found between TAPS and SRS/BrQ for females, previous work showed female body image-QoL associations at baseline through 12 months [33]. Conversely, male patients demonstrated strong positive correlations between TAPS scores and SRS-22r self-image domains at the third evaluation. This observation gains support from validity studies showing visual tools (TAPS drawings r = −0.51; SAQ Appearance r = 0.55) correlate more strongly with spinal deformity than verbal measures (G-BIDQ-S r = 0.30; G-QLPSD r = 0.28) [47–50], suggesting visual assessments better capture scoliosis' immediate body image impact before these perceptions integrate into broader self-evaluation and long-term QoL.

### Limitations

One limitation of the study is the relatively small sample size and the lower number of male participants compared to females, which reflects the approximate 10:1 female-to-male ratio in AIS cases in Hong Kong [51]. This small sample size requires a cautious interpretation of the findings. Additionally, although some patients had electric monitors embedded in their orthoses, most did not, so treatment compliance was self-reported, potentially introducing recall bias. Future studies should incorporate objective monitoring tools. Finally, uneven follow-up periods and the need for longer follow-up restrict the ability to fully understand the long-term effects of orthotic treatment on QoL in the patients with AIS.

### Conclusions

In conclusion, no significant gender differences were obsesved in trunk appearance perception and HRQoL during the orthotic treatment for AIS. Female patients demonstrated progressively poorer body image perception with increasing age, while males showed an association between higher compliance and more negative self-image. Both genders exhibited persistent self-esteem deficits, highlighting the need for integrated psychological support in orthotic management.

## Supporting information

**S1 File. The trunk appearance perception scale.**
(DOCX)

**S2 File. The Scoliosis Research Society-22r questionnaire.**
(DOCX)

**S3 File. The brace questionnaire.**
(DOCX)

## Acknowledgments

Thanks to the participants and their parents for their active participation in this study. Special thanks to the service team in the subject recruitment and management process.

## Author contributions

**Conceptualization:** JoJo Yiying ZOU, Tung LI, Man Sang WONG.

**Data curation:** JoJo Yiying ZOU, Tung LI.

**Formal analysis:** JoJo Yiying ZOU, Tung LI.

**Investigation:** JoJo Yiying ZOU, Tung LI, Man Sang WONG.

**Methodology:** JoJo Yiying ZOU, Tung LI, Man Sang WONG.

**Project administration:** JoJo Yiying ZOU, Man Sang WONG.

**Resources:** JoJo Yiying ZOU, Man Sang WONG.

**Software:** JoJo Yiying ZOU.

**Supervision:** JoJo Yiying ZOU, Man Sang WONG.

**Validation:** JoJo Yiying ZOU.

**Visualization:** JoJo Yiying ZOU, Tung LI, Man Sang WONG.

**Writing – original draft:** JoJo Yiying ZOU, Tung LI.

**Writing – review & editing:** JoJo Yiying ZOU, Man Sang WONG.

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
