## [Decision Letter · Decision Letter 0]

Dear Dr. ZOU,

Thank you for submitting your manuscript to PLOS ONE. After careful consideration, we feel that it has merit but does not fully meet PLOS ONE’s publication criteria as it currently stands. Therefore, we invite you to submit a revised version of the manuscript that addresses the points raised during the review process.

We look forward to receiving your revised manuscript.

Kind regards,

Taher Babaee

Academic Editor

PLOS ONE

**Journal Requirements:**

Reviewers' comments:

Reviewer's Responses to Questions

**Comments to the Author**

1. Is the manuscript technically sound, and do the data support the conclusions?

Reviewer #1: Partly

Reviewer #2: Partly

2. Has the statistical analysis been performed appropriately and rigorously?

Reviewer #1: Yes

Reviewer #2: Yes

3. Have the authors made all data underlying the findings in their manuscript fully available?

Reviewer #1: Yes

Reviewer #2: Yes

4. Is the manuscript presented in an intelligible fashion and written in standard English?

Reviewer #1: Yes

Reviewer #2: Yes

**Reviewer #1: ** The study addresses an important topic, and the effort put forth by the authors in exploring the relationship between bracing and quality of life is commendable. However, several aspects of the manuscript require clarification and improvement to enhance its quality and readability.

Abstract:

Lines 5–9: The sentences lack logical flow, which hinders readability. Additionally, the rationale for the research is not adequately justified. Please provide a clearer explanation of the problem being addressed and the study’s significance.

Regarding the TAPS scale: Is it considered a subscale of quality of life (QoL)? If so, the title is misleading as it separates TAPS and QoL into distinct parameters.

Introduction and Methods:

Lines 40–44: The claims made here require appropriate references to support them.

Lines 55–60: There is a contradiction in this section. While you mention that QoL has been evaluated in both genders, you also state that further investigation is needed for males. What specific limitations in previous studies motivated this research? Please elaborate.

Were all patients prescribed the same type of brace, or were there differences? If variations existed, clarify these details and discuss their potential impact.

Please include an image of the brace used in this study for context. Additionally, the current manuscript contains low-quality images. Replace these with high-resolution photos to ensure clarity and readability.

Lines 103–106: Initially, the third evaluation was described as being conducted 7 months after the start of bracing. Later, this timeline was adjusted due to the pandemic. Ensure consistency in reporting evaluation timelines to avoid confusion.

Results:

In the questionnaire results section, organize the data in a table that includes significance values to improve readability and facilitate interpretation of the findings.

Discussion:

Lines 218–219: You discuss a finding that “for females, higher QoL was linked to an improved self-image,” but there is no corresponding result in the manuscript to confirm this claim. Ensure that all discussed findings are supported by results presented in the study.

Throughout the manuscript, some findings are either inconsistently reported or not adequately supported by the results section. For example:

You report no significant difference in TAPS scores between males and females across all evaluations, yet the discussion suggests otherwise. The second evaluation reportedly showed higher TAPS scores compared to the first evaluation, but this is misinterpreted in the discussion as males having better scores than females. This inconsistency must be corrected to ensure the discussion aligns with the reported results.

Ensure that all claims made in the discussion are clearly presented in the results section, complete with statistical significance values.

**Reviewer #2: ** The author’s approach to exploring the correlation between gender and the psychological impact of adolescent idiopathic scoliosis (AIS) in adolescents who have undergone orthotic treatment is a smart and valuable choice. Overall, the manuscript is well-written and effectively highlights how scoliosis can affect the quality of life (QoL) differently across genders. The study used three well-known and validated questionnaires (TAPS, SRS22-r, and BrQ), completed by 39 participants (29 females, 10 males) at three time points: before starting treatment, one month after, and six months after the second visit. By applying appropriate statistical tests, the authors measured correlations between psychological factors and demographic features in each gender.

The findings were noteworthy: Though both genders had issues addressing self-esteem and poor self-image, each showed different sensitivity in each studied sub-domain. Remarkably, the QoL and trunk appearance perception showed a significant correlation in females, unlike males who showed more sensitivity to brace compliance and the duration of treatment.

The manuscript’s structure and clear articulation of research questions are commendable. However, there are a few areas where addressing certain shortcomings could significantly enhance the overall impact of the study.

Introduction

The introduction provides a concise overview of how gender differences can influence psychological parameters. However, it primarily focuses on the notion that females are more sensitive to these issues—a perspective often cited but potentially biased. While this viewpoint is supported by some studies, there is also evidence suggesting that males may be equally or more affected in certain scenarios. To ensure a balanced discussion, it would be beneficial to reference research supporting both perspectives.

Methodology

The timeline for administering the questionnaires—baseline, one month, and seven months—seems logical but would be more robust if supported by references justifying these specific intervals. Additionally, since the study utilized both English and Chinese versions of the questionnaires, it is important to address the reliability and validity of the Chinese versions in the text, along with relevant supporting data. The rationale for using English versions when the Chinese versions may be more culturally appropriate should also be clarified.

The choice to use both the SRS22-r and BrQ questionnaires is somewhat unclear. While BrQ is specifically designed for assessing brace treatment, SRS22-r is more commonly used for surgical cases and is known to have a ceiling effect in certain sub-domains like satisfaction and pain. Clarifying why SRS22-r was used alongside BrQ would help readers understand its inclusion. Additionally, while the BrQ is a suitable choice, the scoring methodology should be described in more detail.

The TAPS questionnaire also raises some concerns. The first and second sets of questions, respectively depicting the body from a back view and in a forward-bending position, are complicated to be answered subjectively by the patients. Alternatively, the Spinal Appearance Questionnaire (SAQ) could be an easier and more comprehensive option.

Another methodological issue is the decision to have participants complete all three questionnaires at the second and third time points, but only the TAPS at baseline. While it is understandable that BrQ could not be completed before brace treatment, the absence of SRS22-r at baseline is less clear. QoL at the start of treatment could significantly influence later outcomes, and the rationale for excluding this data should be explained.

The inclusion criterion of a minimum 8-hour brace compliance also raises questions. Previous studies, such as Weinstein et al., suggest that the minimum compliance for a TLSO brace is 12.9 hours. Without direct literature supporting the 8-hour criterion, its use in this study is questionable.

Finally, while the authors mention using in-brace X-rays to calculate correction, the exact timing of these images should be specified.

Results

The statistical analysis appears robust and well-executed. The data is presented clearly, and the small sample size is appropriately acknowledged as a limitation. However, providing a preliminary sample size calculation to justify the initial selection of 42 participants would add credibility to the study.

In addition, including p-values in all tables would enhance transparency. For example, the absence of p-values in Table 2 could lead to some ambiguity. .

The discussion of limitations is thorough, but one additional factor could be considered: the uneven follow-up periods caused by the COVID-19 pandemic. This might have influenced the results and should be noted.

Discussion

While the discussion addresses an interesting topic, it lacks sufficient supporting evidence and would benefit from a major revision.

For example, the manuscript mentions societal pressures on young females regarding appearance but does not fully explore the topic: considering the sensitive age of adolescence and cultural differences in each society, not only females but also males could be prone to such difficulties. Although these pressures undeniably exist, the statement would be stronger with more supporting evidence. Exploring the role of cultural factors alongside societal pressures could add depth to the discussion.

Similarly, the statement that males are more sensitive to the discomfort caused by braces is interesting but unsupported by measured data, as discomfort was not explicitly included as a parameter in the study.

The discussion also notes that males with more severe scoliosis are more aware of their condition, leading to better compliance and higher QoL. While this is a plausible hypothesis, it would benefit from additional evidence. Studies suggesting that fear of invasive treatments may drive compliance in severe cases could further support this point.

Lastly, the manuscript could delve into the impact of curve patterns on gender differences. Understanding how specific scoliosis patterns affect QoL in males and females differently could provide valuable insights.

Conclusion

The manuscript effectively tackles an important topic, particularly the psychological aspects of scoliosis and their gender-specific impacts. However, incorporating more evidence, clarifying methodological choices, and revising the discussion section majorly would significantly enhance its quality and impact.

**Do you want your identity to be public for this peer review?** For information about this choice, including consent withdrawal, please see our Privacy Policy

Reviewer #1: **Yes: ** Shahrbanoo Bidari

Reviewer #2: **Yes: ** Yalda Khoshhal

---

## [Author Response · Author response to Decision Letter 1]

17 Feb 2025

Dear Sir/Madam,

Thank you for the opportunity to revise this manuscript and for the constructive comments provided by the reviewers. We appreciate the time and effort invested in evaluating this work. Below, please find our point-by-point responses and the corresponding changes made in the revised manuscript.

Reviewer #1: The study addresses an important topic, and the effort put forth by the authors in exploring the relationship between bracing and quality of life is commendable. However, several aspects of the manuscript require clarification and improvement to enhance its quality and readability.

Abstract:

Comment: Lines 5–9: The sentences lack logical flow, which hinders readability. Additionally, the rationale for the research is not adequately justified. Please provide a clearer explanation of the problem being addressed and the study’s significance.

Response: Thank you for your suggestions. We have revised these lines to clearly explain the problem being addressed and to emphasize the study’s significance. In the abstract section, lines 3-8:

“Although prior studies have evaluated various factors influencing quality of life (QoL) in the patients with AIS, the association between trunk appearance perception and QoL, including potential gender differences, remains insufficiently defined. This study investigated gender differences in trunk appearance perception and its relationship with QoL among the patients with moderate AIS undergoing orthotic treatment.”

Comment: Regarding the TAPS scale: Is it considered a subscale of quality of life (QoL)? If so, the title is misleading as it separates TAPS and QoL into distinct parameters.

Response: Thank you for your comments. TAPS is specifically designed to assess the subjective perception of trunk appearance, which is a feeling of body shape/image. Although body image and self-perception can influence psychological well-being and overall QoL, TAPS is not a subscale of any multifaceted QoL instrument such as the SRS-22r. Instead, it represents an independent measure within the broader context of psychosocial factors that may relate to QoL. To avoid any potential ambiguity, we modified the title so that it clearly indicates that trunk appearance perception (as measured by TAPS) and QoL are independent parameters examined for their association rather than any components of one another:

“Gender differences in trunk appearance perception and its association with quality of life in the patients with moderate adolescent idiopathic scoliosis undergoing orthotic treatment: An observational study”

Clarifications were also added in the introduction section, lines 52-60:

“QoL is a multidimensional concept that includes physical health, functional ability, emotional well-being, and social relationships [1-3]. In the patients with moderate AIS, both spinal deformity and long-term orthotic treatment can lead to psychosocial challenges [4-6]. In particular, trunk appearance perception, how people subjectively evaluate the aesthetic qualities of their torso, plays a key role in overall body image. Changes in trunk appearance can affect self-esteem and emotional health, thereby influencing social interactions and overall QoL [7]. Moreover, prolonged orthosis use can alter lifestyle and increase stress, anxiety, and depression, further reducing self-esteem and leading to a decline in overall QoL [8-10].”

Introduction and Methods:

Comment: Lines 40–44: The claims made here require appropriate references to support them.

Response: References have been added to address the comment in the introduction section, lines 46-49:

“However, the success of orthotic treatment depends not only on its mechanical effectiveness but also on patient compliance, which can be challenging due to the physical discomfort, altered body appearance, and restrictions on daily activities imposed by orthosis wear [11, 12].”

Comment: Lines 55–60: There is a contradiction in this section. While you mention that QoL has been evaluated in both genders, you also state that further investigation is needed for males. What specific limitations in previous studies motivated this research? Please elaborate.

Response: Thank you for highlighting the contradiction and requesting further clarification. We have revised the text to explicitly state the limitations in the previous studies and clarify the motivation for our research, in the introduction section, lines 63-74:

“Current research highlights gender differences in these domains. Females with AIS tend to be more sensitive to their physical appearance and are more likely to experience body dissatisfaction, which is associated with lower QoL, particularly in areas related to body image and social functioning [4, 13]. Studies comparing overall body image perception and QoL between genders have produced mixed results: some have found no significant differences in stress levels, perception of spinal appearance, and QoL [6, 14], while others report that females exhibit lower body image perception and QoL during orthotic treatment [4, 13]. Moreover, there is limited research directly examining the relationship between trunk appearance perception and QoL in males versus females with AIS undergoing orthotic treatment. Although males also face appearance-related challenges, the nature and impact of these issues on their QoL may differ and require further analysis.”

Comment: Were all patients prescribed the same type of brace, or were there differences? If variations existed, clarify these details and discuss their potential impact.

Response: The same type of brace was prescribed for the patients. We revised the description for clarification, in the materials and methods section, lines 93-97:

“Each patient was prescribed a custom-fabricated Hong Kong style TLSO (Figure 1) by a qualified orthotist and instructed to wear it 23 hours a day. The orthosis was equipped with strap markings to ensure correct tightness. All patients received the same type of orthosis, thereby standardizing the treatment and minimizing potential variations in outcomes attributable to differences in orthosis design.”

Comment: Please include an image of the brace used in this study for context. Additionally, the current manuscript contains low-quality images. Replace these with high-resolution photos to ensure clarity and readability.

Response: Thank you for your suggestion. We have now included an image of the brace used in the study and replaced the low-resolution images with high-quality photos throughout the manuscript.

Comment: Lines 103–106: Initially, the third evaluation was described as being conducted 7 months after the start of bracing. Later, this timeline was adjusted due to the pandemic. Ensure consistency in reporting evaluation timelines to avoid confusion.

Response: The sentence was revised for clarification, in the materials and methods section, lines 123-130:

“The third evaluation was originally scheduled for six months after the second assessment (i.e., seven months post-orthosis fitting) to capture longer-term treatment effects [15]. An investigator provided explanations or clarifications as necessary. By the third evaluation, the patients were informed of their in-brace correction results. However, because of disruptions caused by the COVID-19 pandemic, some follow-up sessions were rescheduled to ensure participant safety and data collection, resulting in the third evaluation being completed between 6 and 17 months after having orthotic treatment, with an average follow-up period of 8.9 months.”

Results:

Comment: In the questionnaire results section, organize the data in a table that includes significance values to improve readability and facilitate interpretation of the findings.

Response: Thank you for your suggestion, the p-values were added in all results.

Discussion:

Comment: Lines 218–219: You discuss a finding that “for females, higher QoL was linked to an improved self-image,” but there is no corresponding result in the manuscript to confirm this claim. Ensure that all discussed findings are supported by results presented in the study.

Response: Thank you for your valuable comment. We have revised the discussion by rephrasing the statement to accurately reflect our findings.

Comment: Throughout the manuscript, some findings are either inconsistently reported or not adequately supported by the results section. For example:

You report no significant difference in TAPS scores between males and females across all evaluations, yet the discussion suggests otherwise. The second evaluation reportedly showed higher TAPS scores compared to the first evaluation, but this is misinterpreted in the discussion as males having better scores than females. This inconsistency must be corrected to ensure the discussion aligns with the reported results.

Ensure that all claims made in the discussion are clearly presented in the results section, complete with statistical significance values.

Response: Thank you for your valuable comment. We have carefully reexamined the discussion to ensure that every interpretation is supported by our results. Based on your feedback, we have revised the discussion section to directly reflect our data without extrapolating beyond our findings.

Reviewer #2: The author’s approach to exploring the correlation between gender and the psychological impact of adolescent idiopathic scoliosis (AIS) in adolescents who have undergone orthotic treatment is a smart and valuable choice. Overall, the manuscript is well-written and effectively highlights how scoliosis can affect the quality of life (QoL) differently across genders. The study used three well-known and validated questionnaires (TAPS, SRS22-r, and BrQ), completed by 39 participants (29 females, 10 males) at three time points: before starting treatment, one month after, and six months after the second visit. By applying appropriate statistical tests, the authors measured correlations between psychological factors and demographic features in each gender.

The findings were noteworthy: Though both genders had issues addressing self-esteem and poor self-image, each showed different sensitivity in each studied sub-domain. Remarkably, the QoL and trunk appearance perception showed a significant correlation in females, unlike males who showed more sensitivity to brace compliance and the duration of treatment.

The manuscript’s structure and clear articulation of research questions are commendable. However, there are a few areas where addressing certain shortcomings could significantly enhance the overall impact of the study.

Introduction

Comment: The introduction provides a concise overview of how gender differences can influence psychological parameters. However, it primarily focuses on the notion that females are more sensitive to these issues—a perspective often cited but potentially biased. While this viewpoint is supported by some studies, there is also evidence suggesting that males may be equally or more affected in certain scenarios. To ensure a balanced discussion, it would be beneficial to reference research supporting both perspectives.

Response: Thank you for your thoughtful comment regarding the potential bias in our introduction. We agree that while many studies highlight that females with AIS tend to exhibit greater sensitivity to body image issues, there are also some studies suggesting that no significant difference was found between males and females. In response, we have revised our introduction to incorporate a more balanced discussion.

Methodology

Comment: The timeline for administering the questionnaires—baseline, one month, and seven months—seems logical but would be more robust if supported by references justifying these specific intervals.

Response: Relevant references were added to address this comment, in the materials and methods section, lines 117-125:

“The evaluation timeline was based on clinical practice guidelines and some previous studies. The first evaluation, conducted before orthosis fitting, employed only the Trunk Appearance Perception Scale (TAPS) to capture baseline trunk appearance perception. One month after the start of orthotic treatment, an integrated questionnaire, comprising TAPS, the Scoliosis Research Society-22r (SRS-22r), and the Brace questionnaire (BrQ), was administered to assess early adaptation, allowing patients to adjust their lifestyle to the orthosis [16]. The third evaluation was originally scheduled for six months after the second assessment (i.e., seven months post-orthosis fitting) to capture longer-term treatment effects [15].”

Comment: Additionally, since the study utilized both English and Chinese versions of the questionnaires, it is important to address the reliability and validity of the Chinese versions in the text, along with relevant supporting data. The rationale for using English versions when the Chinese versions may be more culturally appropriate should also be clarified.

Response: Relevant references were added to support the use of Chinese-translated questionnaires, in the materials and methods section:

Lines 146-148:

“Similarly, the Chinese-translated TAPS has been validated in prior studies, showing acceptable internal consistency (Cronbach’s α = 0.70) and reproducibility (ICC = 0.95) [17].”

Lines 158-159:

“Moreover, the Chinese-translated SRS-22 questionnaire has shown excellent test-retest reproducibility, with an ICC of 0.75 across all domains [18].”

Lines 169-172:

“The original BrQ demonstrates satisfactory internal consistency reliability (Cronbach's alpha = 0.82) [19] and the Chinese-translated version has shown very good internal consistency (Cronbach's alpha = 0.89) and excellent test-retest reproducibility (ICC = 0.83)[20].”

In this study, we used both the English and Chinese versions of the questionnaires to accommodate the bilingual nature in Hong Kong. This dual-language approach enables direct comparisons with international studies while allowing participants to select their preferred language, thereby enhancing comprehension and data quality. Revision was added in the manuscript to address the comment, in the materials and methods section, lines 132-135:

“Three questionnaires were used in this study, with both English and Chinese versions available for patients in a bilingual society. This dual-language approach facilitates direct comparisons with international studies and ensures that participants can select their preferred language, enhancing comprehension and data quality.”

Comment: The choice to use both the SRS22-r and BrQ questionnaires is somewhat unclear. While BrQ is specifically designed for assessing brace treatment, SRS22-r is more commonly used for surgical cases and is known to have a ceiling effect in certain sub-domains like satisfaction and pain. Clarifying why SRS22-r was used alongside BrQ would help readers understand its inclusion. Additionally, while the BrQ is a suitable choice, the scoring methodology should be described in more detail.

Response: We used the BrQ because it is designed specifically to assess the patient experience during orthotic management, while the SRS22-r is widely used in scoliosis management, allowing us to compare our results with previous studies. Despite its ceiling effects in some areas, the SRS22-r provides general QoL aspects that complement the information from the BrQ, in the materials and methods section, lines 150-153:

“The SRS-22r questionnaire, recommended by the Scoliosis Research Society (SRS) for evaluating treatment outcomes, assesses health-related QoL in the patients with AIS and has been widely used in previous research as a comprehensive measure of overall QoL [4, 21-23].”

The scoring methodology was further explained in the manuscript, lines 163-168:

“Each item is rated on a 5-point Likert scale from 1 (poorest) to 5 (best). The total BrQ score is obtained by summing all 34 item scores, multiplying the sum by 20, and dividing by 34, yielding a normalized score between 20 and 100 (with higher scores indicating better QoL). A subscale score for each domain is calculated by dividing the domain’s total score by the number of items in that domain.”

Comment: The TAPS questionnaire also raises some concerns. The first and second sets of qu

---

## [Decision Letter · Decision Letter 1]

Dear Dr. ZOU,

Thank you for submitting your manuscript to PLOS ONE. After careful consideration, we feel that it has merit but does not fully meet PLOS ONE’s publication criteria as it currently stands. Therefore, we invite you to submit a revised version of the manuscript that addresses the points raised during the review process.

We look forward to receiving your revised manuscript.

Kind regards,

Taher Babaee

Academic Editor

PLOS ONE

Reviewers' comments:

Reviewer's Responses to Questions

**Comments to the Author**

Reviewer #1: All comments have been addressed

Reviewer #2: (No Response)

2. Is the manuscript technically sound, and do the data support the conclusions?

Reviewer #1: Yes

Reviewer #2: Yes

3. Has the statistical analysis been performed appropriately and rigorously?

Reviewer #1: Yes

Reviewer #2: Yes

4. Have the authors made all data underlying the findings in their manuscript fully available?

Reviewer #1: Yes

Reviewer #2: Yes

5. Is the manuscript presented in an intelligible fashion and written in standard English?

Reviewer #1: Yes

Reviewer #2: Yes

Reviewer #1: The revisions were addressed. I think the manuscript can be published after the final confirmation from the journal.

Reviewer #2: The Authors have addressed the majority of comments from the first review which have indeed resulted in a more comprehensive and clear manuscript but unfortunately some critical questions remain unanswered.

The structure of the Introduction has undoubtedly been improved and the importance of the topic is effectively articulated, nonetheless the most important matter in this section is still unresolved. It is clear that the authors have tried to demonstrate that both genders are sensitive to the issue but the initial bias about the higher sensitivity of females is still remarkably evident. This matter should either be with more literature supported or better discussed.

The methodology section raised many questions in the first review, and the majority of them are, sadly, still unanswered. Validating the chosen timeline for assessments with previous studies is, though, a positive approach, however, the choice of supporting evidence is not robust. Furthermore, the minimum of 8-hour brace compliance is as before unjustified. On the other hand, neither the reason for using both SRS22 and BrQ together nor the explanation for the absence of SRS-22r at baseline has been clearly discussed.

Turning to the discussion and conclusion, it should be mentioned that the manuscript is more solid and more effectively aligned with the results, but sadly the findings are not adequately discussed and legitimated. Although most of the comments are considered and applied, unfortunately the lack of enough supporting evidence is still undeniable.

Considering all mentioned above, despite the crucial matter and valuable findings, the paper should be better written and justified to be more thoroughly understood by the reader.

**Do you want your identity to be public for this peer review?** For information about this choice, including consent withdrawal, please see our Privacy Policy

Reviewer #1: **Yes: ** Shahrbanoo Bidari

Reviewer #2: **Yes: ** Yalda Khoshhal

---

## [Author Response · Author response to Decision Letter 2]

26 Apr 2025

Dear Editor,

Thank you for the opportunity to revise our manuscript. During the revision process, we were able to collect and incorporate data from six additional participants, bringing the total sample size to 45. This enhancement allowed us to increase representation in both female and male subgroups. In addition, we also included age as a variable for further analysis.

We are pleased to submit the updated manuscript, which maintains all original key conclusions: No significant gender differences in trunk appearance perception or health-related quality of life (HRQoL). The self-image domain remains the lowest-scoring among all HRQoL domains.

We would appreciate your guidance on whether this revised approach is acceptable.

Please let us know if any further clarification or adjustments are needed.

Kind regards,

JoJo

Reviewer #1: The revisions were addressed. I think the manuscript can be published after the final confirmation from the journal.

Response: Thank you for confirming the revisions. We appreciate the opportunity to improve our manuscript.

Reviewer #2: The Authors have addressed the majority of comments from the first review which have indeed resulted in a more comprehensive and clear manuscript but unfortunately some critical questions remain unanswered. The structure of the Introduction has undoubtedly been improved and the importance of the topic is effectively articulated, nonetheless the most important matter in this section is still unresolved. It is clear that the authors have tried to demonstrate that both genders are sensitive to the issue but the initial bias about the higher sensitivity of females is still remarkably evident. This matter should either be with more literature supported or better discussed.

Response: Thank you for the constructive feedback. We have adjusted the introduction section in response.

Comment: The methodology section raised many questions in the first review, and the majority of them are, sadly, still unanswered. Validating the chosen timeline for assessments with previous studies is, though, a positive approach, however, the choice of supporting evidence is not robust.

Response: Thank you for your comments. We further explained the reason of the decision of the evaluation timeline in the method section, lines 150 – 161:

“The evaluation timeline was based on clinical practice and some previous studies . The first evaluation, conducted before orthosis fitting, employed only the TAPS to capture baseline trunk appearance perception [33, 34]. One month after treatment initiation, an integrated questionnaire (TAPS, SRS-22r, and BrQ) was administered to assess early adaptation. This one-month follow-up was strategically chosen because the initial orthotic treatment period is critical for long-term outcomes [35], while also aligning with standard clinical protocols that allow sufficient time for patient adaptation yet enable early orthosis adjustments if needed. The third evaluation was originally scheduled for six months after the second assessment (i.e., seven months post-orthosis fitting) to capture longer-term treatment effects, in accordance with both the 2018 SOSORT guidelines recommending 3 – 6 months follow-ups [36] and protocols from previous studies using 6-month intervals [17, 33].”

Comment: Furthermore, the minimum of 8-hour brace compliance is as before unjustified.

Response: Thank you for your comments. After careful consideration, we have removed the minimum requirement of 8-hour daily brace compliance from our inclusion criteria. Recruit patients with high compliance could introduce a bias, as it may indicate a more positive attitude to the orthotic treatment. We intended to capture a whole range of treatment behaviors and evaluate the dose-response relationship between wear time and QoL. This concern was also addressed in our initial review response.

Comment: On the other hand, neither the reason for using both SRS22 and BrQ together nor the explanation for the absence of SRS-22r at baseline has been clearly discussed.

Response: The SRS-22r is a standard clinical questionnaire that has been widely used in previous studies, allowing for direct comparisons of our results with the literature and followers. Additionally, since the SRS-22r questionnaire is used to assess the experience gone through during intervention, it is not administered at baseline. This was also mentioned in our initial review response.

Comment: Turning to the discussion and conclusion, it should be mentioned that the manuscript is more solid and more effectively aligned with the results, but sadly the findings are not adequately discussed and legitimated. Although most of the comments are considered and applied, unfortunately the lack of enough supporting evidence is still undeniable.

Response: Thank you for the constructive feedback. We have revised the discussion section in response.

Considering all mentioned above, despite the crucial matter and valuable findings, the paper should be better written and justified to be more thoroughly understood by the reader.

---

## [Decision Letter · Decision Letter 2]

Gender differences in trunk appearance perception and health-related quality of life (HRQoL) in the patients with moderate adolescent idiopathic scoliosis (AIS) undergoing orthotic treatment: An observational study

PONE-D-24-49341R2

Dear Dr. ZOU,

We’re pleased to inform you that your manuscript has been judged scientifically suitable for publication and will be formally accepted for publication once it meets all outstanding technical requirements.

Kind regards,

Taher Babaee

Academic Editor

PLOS ONE

Additional Editor Comments (optional):

Reviewers' comments:

Reviewer's Responses to Questions

**Comments to the Author**

Reviewer #2: All comments have been addressed

2. Is the manuscript technically sound, and do the data support the conclusions?

Reviewer #2: Yes

3. Has the statistical analysis been performed appropriately and rigorously?

Reviewer #2: Yes

4. Have the authors made all data underlying the findings in their manuscript fully available?

Reviewer #2: Yes

5. Is the manuscript presented in an intelligible fashion and written in standard English?

Reviewer #2: Yes

Reviewer #2: (No Response)

**Do you want your identity to be public for this peer review?** For information about this choice, including consent withdrawal, please see our Privacy Policy

Reviewer #2: **Yes: ** Yalda Khoshhal

---

## [Editor Report · Acceptance letter]

PONE-D-24-49341R2

PLOS ONE

Dear Dr. ZOU,

I'm pleased to inform you that your manuscript has been deemed suitable for publication in PLOS ONE. Congratulations! Your manuscript is now being handed over to our production team.

Kind regards,

on behalf of

Dr. Taher Babaee

Academic Editor

PLOS ONE